# Hysteresis Modeling and Compensation of Fast Steering Mirrors with Hysteresis Operator Based Back Propagation Neural Networks

**DOI:** 10.3390/mi12070732

**Published:** 2021-06-22

**Authors:** Kairui Cao, Guanglu Hao, Qingfeng Liu, Liying Tan, Jing Ma

**Affiliations:** School of Astronautics, Harbin Institute of Technology, Harbin 150001, China; cao-kairui@hit.edu.cn (K.C.); alwayslqf@163.com (Q.L.); tanliying@hit.edu.cn (L.T.); majing@hit.edu.cn (J.M.)

**Keywords:** nonlinear hysteresis, inverse hysteresis compensation, fast steering mirror (FSM), neural network, Madelung’s rules

## Abstract

Fast steering mirrors (FSMs), driven by piezoelectric ceramics, are usually used as actuators for high-precision beam control. A FSM generally contains four ceramics that are distributed in a crisscross pattern. The cooperative movement of the two ceramics along one radial direction generates the deflection of the FSM in the same orientation. Unlike the hysteresis nonlinearity of a single piezoelectric ceramic, which is symmetric or asymmetric, the FSM exhibits complex hysteresis characteristics. In this paper, a systematic way of modeling the hysteresis nonlinearity of FSMs is proposed using a Madelung’s rules based symmetric hysteresis operator with a cascaded neural network. The hysteresis operator provides a basic hysteresis motion for the FSM. The neural network modifies the basic hysteresis motion to accurately describe the hysteresis nonlinearity of FSMs. The wiping-out and congruency properties of the proposed method are also analyzed. Moreover, the inverse hysteresis model is constructed to reduce the hysteresis nonlinearity of FSMs. The effectiveness of the presented model is validated by experimental results.

## 1. Introduction

Based on the advantages of fast response, high stiffness, and high electromechanical coupling efficiency, fast steering mirrors (FSMs), driven by piezoelectric ceramics, are widely used as actuators in the field of beam control [1,2,3,4]. FSM has a very precise optical and mechanical structure. In a FSM, four piezoelectric ceramics are symmetrically distributed in a crisscross pattern. The mirror platform is connected to the housing by flexure hinges. These four piezoelectric ceramics cooperate to push the mirror platform to deflect. The motion of each two ceramics along one radial direction generates the deflection of the FSM in the same orientation. As shown in Figure 1, the opposing two piezoelectric ceramics in one direction are connected in series to share a fixed voltage of 100 V. The control voltage is injected into to the electrical junction of these two ceramics. Changing the control voltage is equal to varying the voltages imposed on the ceramics. The ceramics are elongated or shortened according to the change of the control voltage. Then the mirror platform is deflected by the motions of the ceramics.

It is known that piezoelectric ceramics have a strong hysteresis effect. Many efforts have been devoted to describe the hysteresis nonlinearity of piezoelectric ceramics. The well-known Preisach model [5], Prandt–Ishlinskii (PI) model [6], Maxwell model [7,8], and Bouc–Wen model [9] are hysteresis models. Furthermore, modified versions of these models have been developed to describe the asymmetric hysteresis nonlinearity [10,11,12,13,14,15,16,17].

Unlike a single piezoelectric ceramic, the relationship between input voltage and output angle of the FSM shows that the hysteresis nonlinearity is complex, which cannot be accurately described by the above-mentioned hysteresis models. It is valuable to develop new hysteresis description methods. In [18,19], neural networks are introduced to describe the hysteresis nonlinearity of FSMs, where the multi-valued mapping of the hysteresis is transformed into one-to-one mapping by neural networks. However, more complex hysteresis nonlinearity means more neurons are needed in neural networks.

Inspired by [20], the structure of the hysteresis model proposed in this paper consists of a hysteresis operator and a cascaded back propagation (BP) neural network. In our method, the hysteresis operator provides basic hysteresis motions for FSMs, and the neural network modifies the basic hysteresis motion to accurately describe the hysteresis nonlinearity of FSMs. However, it should be emphasized that the results in [20] cannot be generalized for FSMs directly since the hysteresis operator of [20] is not complete. The focus of this paper is on the modeling and compensation of the hysteresis nonlinearity for FSMs, and the main contributions of this paper are as follows: (1) The proposed hysteresis operator based on Madelung’s rules is an important extension of the hysteresis operator in [20]. In the presented hysteresis operator, we provide a concrete realization algorithm that can be directly used in practice. (2) We propose a systematic way to describe and compensate the hysteresis nonlinearity of FSMs. Furthermore, the wiping-out and congruency properties of the proposed hysteresis model are investigated. (3) The inverse hysteresis model is constructed to reduce the hysteresis effect of FSMs.

The remainder of this paper is organized as follows. In Section 2, the structure of the proposed hysteresis modeling method for FSMs is elaborated in detail. In Section 3, the properties of the presented method are investigated and validated by simulations. Experiments are conducted in Section 4 to show the effectiveness of the presented modeling and compensation methods. Some remarks and conclusions are drawn in Section 5.

## 2. Hysteresis Model

The structure of the proposed hysteresis model is shown in Figure 2, where a symmetric hysteresis operator with a cascaded BP neural network is used to describe the hysteresis nonlinearity of FSMs. The hysteresis operator is a symmetric hysteresis model, which provides a basic hysteresis movement for the whole model. Next, the output of the hysteresis operator is modified by the BP neural network to further promote the description accuracy. The details of the model are elaborated in the following subsections.

### 2.1. Hysteresis Operator Based on Madelung’s Rules

#### 2.1.1. Madelung’s Rules and Its Application in Finding the CTP (Current Turning Point)

In this subsection, a symmetric hysteresis operator based on Madelung’s rules is presented. In our method, the information of the major ascending curve is used to describe the inner hysteresis curves. Before proceeding, Madelung’s rules [21,22] are introduced in Figure 3.

The rules of describing hysteresis phenomena were proposed by German physicist Madelung in the early 20th century. The first and second rules describe the properties of hysteresis curves, and the third rule is the well-known wiping-out property. In fact, Madelung’s rules are usually used as criteria to validate newly developed hysteresis models [23,24]. It is noted that the input-output trajectory at any time must be on some hysteresis curve, and this curve is denoted as the current hysteresis curve (CHC). The starting point of the CHC is named the current turning point (CTP), and the CHS is exactly the hysteresis curve between the CTP and the turning point previous the CTP. According to the first rule of Madelung, the CHC can be determined if the CTP is known previously. Therefore, the description of the hysteresis nonlinearity boils down to finding the CTP first, and then describing the CHC.

Next, we will show how to use the wiping-out property to find the CTP through an simple example. As shown in Figure 4, we assume that turning points $A_{1},A_{2},\cdots ,A_{m}$ are generated in sequence under some input signal (see Figure 3). As shown in Figure 4, each local extreme point of the input signal is one-to-one mapping with a turning point. According to the wiping-out property, a traverse event happens when the input signal surpasses its previous extreme points. For instance, as shown in Figure 3 and Figure 4, the CTP becomes $A_{m-2}$ after the traverse, and $A_{m-1}$ and $A_{m}$ should be wiped out since they are no longer involved in the description of the CHC. In Figure 3 and Figure 4, a turning point can further be classified into two types: left turning point and right turning point. If the x-coordinates of the left and right turning points are written into left stack $X_{L}$ and right stack $X_{R}$, respectively, we can see that both arrays are naturally sorted in order of size (see Figure 4). When a new turning point is generated, we can add its x-coordinate at the end of the corresponding stack directly according to its type.

Generally, we cannot predict how many turning points will be traversed for an arbitrary input value x; therefore, we have to compare the input value with all the existing turning points. In order to speed up the comparison, the binary search is adopted. If the CHC is some ascending hysteresis curve, the CTP is a left turning point. At this time, by locating the position of the input value in the right stack with binary search, we will know how many right turning points have been traversed by the trajectory. After wiping out these right turning points and their corresponding left turning points, the CTP can be determined, as shown in Figure 5 where $x_{3}<x<x_{1}$. The same analysis method is also applicable for the case that the CTP is a right turning point.

#### 2.1.2. Description of Hysteresis Curves

In this subsection, the functions of hysteresis curves are derived using the symmetry property. Assume that $f_{A_{0}{A^{\prime }}_{0}}(x)=f_{0}(x)$ is the function of the ascending curve of the major hysteresis loop as shown in Figure 3, where
(1)$$f_{0}(x)=\sum_{i=1}^{n}a_{i}x^{n},\;x\in [x_{0},\;{x^{\prime }}_{0}]$$

In Equation (1), the coefficients $a_{i}$ (i=1, ⋯, n and n is the order) are parameters that need to be identified. According to the symmetry property, the function of the major descending curve $\overset{⏜}{{A^{\prime }}_{0}A_{0}}$ is as follows:(2)$$f_{{A^{\prime }}_{0}A_{0}}(x)={y^{\prime }}_{0}+y_{0}-f_{0}(x_{0}+{x^{\prime }}_{0}-x),\;x\in [x_{0},\;{x^{\prime }}_{0}]$$

Geometrically, $f_{A_{0}{A^{\prime }}_{0}}(x)$ and $f_{{A^{\prime }}_{0}A_{0}}(x)$ are symmetrical with the center point of the major hysteresis loop. Using the symmetry property again, the functions $f_{A_{1}A_{0}}(x)$ and $f_{A_{2}A_{1}}(x)$ of the curves $\overset{⏜}{A_{1}A_{0}}$ and $\overset{⏜}{A_{2}A_{1}}$ can be expressed as
(3)$$f_{A_{1}A_{0}}(x)=y_{1}+y_{0}-f_{0}(x_{0}+x_{1}-x),\;x\in [x_{0},\;x_{1}]$$
(4)$$f_{A_{2}A_{1}}(x)=y_{2}-y_{0}+f_{0}(x_{0}-x_{2}+x),\;x\in [x_{1},\;x_{2}]$$

In fact, $f_{A_{1}A_{0}}(x)$ is the parallel shift of $f_{{A^{\prime }}_{0}A_{0}}(x)$ from ${A^{\prime }}_{0}$ to $A_{1}$. Based on the property of parallel shift, we have
(5)$$y_{1}-f_{A_{1}A_{0}}(x)={y^{\prime }}_{0}-f_{{A^{\prime }}_{0}A_{0}}({x^{\prime }}_{0}-x_{1}+x),\;x\in [x_{0},\;x_{1}]$$

Substituting Equation (2) into Equation (5), we can obtain that the expression of $f_{A_{1}A_{0}}(x)$, which is exactly the same as Equation (3). Similarly, $f_{A_{2}A_{1}}(x)$ (see Equation (4)) can also be derived by the translation of $f_{A_{0}{A^{\prime }}_{0}}(x)$ from $A_{0}$ to $A_{2}$. This fact means that any hysteresis curve can be described by the translation of the major curves. For instance, as shown in Figure 3, the hysteresis curves $\overset{⏜}{A_{m}A_{m-1}}$ and $\overset{⏜}{A_{m-1}A_{m}}$ can be derived by the translation of parts of the ascending and descending curves (see the red lines), respectively. Mathematically, the function of the hysteresis curve $\overset{⏜}{A_{k}A_{k-1}}$ can be written as
(6)$$f_{A_{k}A_{k-1}}(x)=\left\{\begin{array}{l} y_{k}+y_{0}-f_{0}(x_{0}+x_{k}-x)\;\mathrm{for}\;k\;\mathrm{is}\;\mathrm{odd},\;,\;x\in [x_{k-1},\;x_{k}]\\ y_{k}-y_{0}+f_{0}(x_{0}-x_{k}+x)\;\mathrm{for}\;k\;\mathrm{is}\;\mathrm{even},\;x\in [x_{k},\;x_{k-1}] \end{array}\right.$$
where k=1, ⋯, m. It can be seen from Equation (6) that the function of $\overset{⏜}{A_{k}A_{k-1}}$ is determined by $A_{k}$, which validates the first rule of Madelung.

Based on the above analysis, the realization algorithm of the hysteresis operator is summarized in Figure 6. In the algorithm, x is the input value. Auxiliary variables $x_{1}$ and $x_{2}$ are two input values before x, and the inequality $(x-x_{1})(x_{1}-x_{2})<0$ is used to identify whether $\left(x_{1},\;y_{1}\right)$ is a turning point. Stacks $X_{L}$ and $Y_{L}$ (or $X_{R}$ and $Y_{R}$) are used to store all the left turning points (or right turning points). The variable TP_flag denotes the type of the latest turning point. For example, if $(x-x_{1})(x_{1}-x_{2})<0$ and $x-x_{1}>0$, $\left(x_{1},\;y_{1}\right)$ is a left turning point and TP_flag is set as 0. Then, the CTP can be determined by locating x in $X_{L}$ (or $X_{R}$) and all the turning points after the CTP need to be wiped out to save the memory resources. Using Equation (6), the output of the hysteresis operator can be computed. It should be emphasized that the update is only performed when $x\neq x_{1}$ since successive identical input values influence the identification of turning points and consequently lead to erroneous results. Compared with the present hysteresis operator, the operator in [20] is not complete and cannot be directly used in practice, where there is no mechanism to guarantee the wiping-out property.

### 2.2. BP Neural Network

In Figure 2, a three-layer BP neural network is applied to modify the output of the hysteresis operator. The neural network has two inputs and one output. The hidden layer and output layer adopt the sigmoid function and the linear function, respectively. The output of the neural network is as follows:(7)$$S[x,f(x)]=\sum_{j=1}^{n_{b}}w_{j}(\sum_{i=1}^{n_{b}}sigmoid(w_{1i}x+w_{2i}f(x)+b_{i})+c)$$
where $n_{b}$ is the number of neurons, sigmoid(⋅) is the traditional Sigmoid function, and $w_{1i},\;w_{2i},\;w_{j},\;b_{i},\;c$ are the weight and bias parameters that need to be identified.

It is noted that the input-output relationship of the hysteresis operator is a multi-valued mapping. For different input values $x_{1}$ and $x_{2}$, their output values may be identical, i.e., y. Scalar functions cannot be used to modify the output of the hysteresis operator. To distinguish the same output values for different input values, both the input and output of the hysteresis operator are needed, as shown in Figure 2. Therefore, the modification function has two inputs and one output. Apparently, it is difficult to describe the modification function by traditional functions, and so the neural network is the best choice. In our method, the BP neural network is applied. As we will see in the following sections, the application of the neural network could largely promote the hysteresis modeling accuracy for FSMs.

## 3. Wiping-Out and Congruency Properties 

### 3.1. Wiping-Out Property

Based on the above analysis, it can be concluded that the proposed hysteresis model possesses the wiping-out property, since the hysteresis operator naturally possesses this property and the BP neural network is memory-less. This fact can be validated by the following simulation example.

In Figure 7a, an input signal with local extremes $\left(0,x_{1},x_{2},x_{3},x_{4},x_{5}\right)$ is used to validate the wiping-out property. The turning points generated by the input signal are shown in Figure 7b, where the parameters of the proposed model are chosen as the identified parameters in the following experiment (see Equation (8) and Table 1). In the beginning, the input x(t) starts from 0 and reaches its first maximum value $x_{1}$. This part of input generates the trajectory from the initial point $A_{0}$ to memory point $A_{1}$, as depicted in Figure 7b. Then the input x(t) decreases to its first minimum $x_{2}$, and the corresponding hysteresis trajectory returns back to $A_{0}$ along the $\overset{⏜}{A_{1}A_{0}}$ curve. Next, x(t) increases again to the second maximum value $x_{3}$. Correspondingly, the trajectory moves from point $A_{2}$ to point $A_{3}$. The wiping-out event is triggered when the input x(t) decreases to the second minimum value $x_{4}$ since $x_{4}<x_{2}$. In the procedure, the trajectory can be divided into two parts: (1) $\overset{⏜}{A_{3}A_{2}}$ for $x_{2}<x<x_{3}$ and (2) $\overset{⏜}{A_{2}A_{4}}$, which is a section of $\overset{⏜}{A_{1}A_{0}}$, for $x_{3}\leq x\leq x_{4}$. In the second part, $A_{2}$ as well as $A_{3}$ should be wiped out because they are not involved in determining the future hysteresis trajectory. Similarly, the wiping-out event is triggered again as the input x(t) increases from $x_{4}$ to $x_{5}$. The trajectory switches from $\overset{⏜}{A_{4}A_{1}}$ to $\overset{⏜}{A_{0}{A^{\prime }}_{0}}$ at $x=x_{1}$. It can be seen that the whole process is not affected by any past extremes of the input because all the memories about the local turning points have been completely wiped out. This example shows that the proposed hysteresis model possesses the wiping out property.

### 3.2. Congruency Property

The congruency property means that the inner hysteresis loops with the same input range are congruent. In this subsection, we will show the hysteresis operator fulfills the congruency property while the proposed hysteresis model does not possess this property. The input x(t) used for validation is depicted in Figure 8, where the extremes are $\left(0,x_{1},x_{2},x_{3},x_{4},x_{5},0\right)$, where $x_{1}=x_{5}$ and $x_{2}=x_{4}$. In this case, the input ranges of the $\left[x_{2},x_{1}\right]$ and $\left[x_{4},x_{5}\right]$ intervals are identical.

Under the input of Figure 8, the input-output trajectory of the hysteresis operator is depicted in Figure 9a, where two inner loops $L_{1}$ and $L_{2}$ are generated and they have the same input range. An enlarged view of Figure 9a is shown in Figure 9b, where we can observe that $L_{1}$ and $L_{2}$ are congruent, which demonstrates that the two minor loops are congruent. The hysteresis trajectory of the proposed model is shown in Figure 10, where it can be seen that the inner loops are not congruent. The above results imply that the presented model could distinguish the differences of the hysteresis trajectory under the same input ranges for different time periods. This fact reveals that the proposed hysteresis model has the ability to describe complex hysteresis nonlinearity.

## 4. Experimental Verification

### 4.1. Experimental Setup

An experiment platform is established to validate the effectiveness of the proposed hysteresis model, as shown in Figure 11. The platform consists of a host computer, a target computer, a FSM (S330, Physik Instrumente), a high voltage driver (E00, Harbin Core Tomorrow), and an AD/DA card (PCI-6221, National Instruments). The PCI-6221 is used to realize 16-bit AD and DA conversions. The high voltage driver E00 has a fixed gain of 20. The strain gauge sensor (SGS) inside the FSM is used to measure the deflection angle. The angle signal is also amplified by the E00 driver, and then sampled by the PCI-6221.

### 4.2. Parameter Identification

The first step before using the presented model is the parameter identification. In this step, the parameters of the hysteresis operator are determined first and then those of the BP neural network are identified.

A 0.5 Hz sine signal with peak-to-peak value 100 V is used to excite the FSM and the input and output signals are collected by the PCI-6221. The normalized hysteresis trajectory of the FSM, i.e., the major hysteresis loop, is shown in Figure 12. It is known that we only need to determine the major ascending curve $f_{0}(x)$ before using the hysteresis operator. Furthermore, the hysteresis operator only provides the basic hysteresis movement in our method. Therefore, for the sake of simplicity, we have used the major ascending curve of the FSM to model the hysteresis operator. Using the least-square method, the major ascending curve $f_{0}(x)$ can be derived as follows:
(8)$$f_{0}(x)=-0.1347x^{5}+0.5444x^{4}-0.7945x^{3}+0.7135x^{2}+0.6712x$$
where the order n of $f_{0}(x)$ is chosen as 5.

Next, a triangle signal with decreasing amplitude is injected to the FSM to identify the parameters of the BP neural network (see Equation (7)). To balance the efficiency and the accuracy, twelve neurons are used in the middle layer of the neural network. According to Equations (6) and (8), the input and output data of the hysteresis operator, which is also the input data of the neural network, can be derived. Using the Levenberg-Marquardt back-propagation algorithm, the parameters of the neural network can be trained. In Equation (7), the value of c is −1.6447 and other parameters are listed in Table 1.

### 4.3. Hysteresis Description Results

To validate the effectiveness of the proposed model, the sine signal in Figure 13a is used to excite the FSM. The actual output angle and predictive angle of our model are depicted in Figure 13b, and error signal is shown in Figure 13c, where the normalized maximum error (NME) $e_{N}$ is
(9)$$e_{N}=\frac{\max\left|y_{ai}-y_{i}\right|}{\max(y_{i})-\min(y_{i})}\times 100\%=1.38\%$$
and the normalized root-mean-square error (NRMSE) $e_{R}$ is
(10)$$e_{R}=\frac{\sqrt{\frac{1}{N}\sum {\left(y_{ai}-y_{i}\right)}^{2}}}{\max(y_{i})-\min(y_{i})}\times 100\%=0.22\%$$
where $y_{ai}$ and $y_{i}$ are actual and predictive angle, respectively, and N is the number of samples. The actual and predictive hysteresis loops are shown in Figure 13d, where we can see that the proposed hysteresis model well describes the hysteresis nonlinearity of the FSM.

Unlike the asymmetric hysteresis nonlinearity of a single piezoelectric ceramic, in Figure 13d we can see that the hysteresis nonlinearity of FSMs does not show convex-concave characteristic. Thus, one may wonder whether it is enough to describe the hysteresis nonlinearity with the symmetric hysteresis operator only. There are five hysteresis loops in Figure 13d, and we depict the differences of each loop in Figure 14, where it is obvious that the description accuracy is largely improved using the BP neural network. This fact could also be validated by the results in Figure 13c, where the NRMSE $e_{R}$ is reduced from 0.62% to 0.22% with the BP neural network.

### 4.4. Hysteresis Compensation Results

To reduce the hysteresis effect, the traditional way is to construct an inverse hysteresis model, as shown in Figure 15, where $y_{d}$ is the desired input and y is the actual output. The inverse of the proposed hysteresis model can be obtained by replacing the input and the output data of the FSM and performing the procedure of parameter identification again. Figure 16a shows the compensation results, where the actual trajectory well follows the desired trajectory. Figure 16b depicts the tracking error, where the NME $e_{N}$ and the NRMSE $e_{R}$ are 2.42% and 0.71%, respectively. Figure 16c depicts the input-output relationships for both the inverse model and the FSM, where it can be seen that the former is a reflection of the later about the 45° line. It means that if the hysteresis nonlinearity of the FSM can be describe by Φ, then the inverse model can be represented by $\Phi^{-1}$, and we have $y=\Phi (\Phi^{-1}(y_{d}))=y_{d}$. Figure 16d illustrates the desired output versus the actual output, where we can see that the hysteresis nonlinearity of the FSM is significantly reduced.

## 5. Conclusions

In this paper, a systematic way to describe the hysteresis nonlinearity of fast steering mirrors has been developed. The proposed hysteresis model consists of a hysteresis operator and a cascaded back propagation neural network. The hysteresis operator, which provides basic hysteresis motions, is developed based on Madelung’s rules. Using the binary search, the current turning point and current hysteresis curve can be effectively determined. Then, the BP neural network further modifies the output of the hysteresis operator to promote the modelling accuracy. The wiping-out and congruency properties of the proposed model are also investigated. The experimental results have shown that the presented model well describes the hysteresis nonlinearity of FSMs, both for major and inner hysteresis loops. Moreover, the effectiveness of the compensation method is also validated by experiments.

## Figures and Tables

**Figure 1 micromachines-12-00732-f001:**
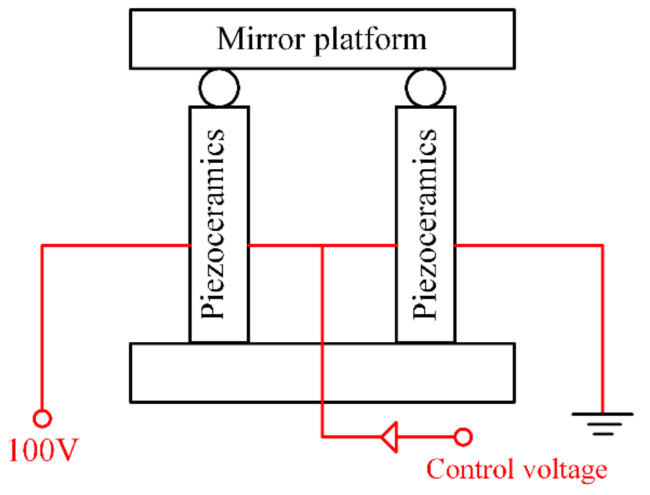
Angle control mechanism of the fast steering mirror.

**Figure 2 micromachines-12-00732-f002:**
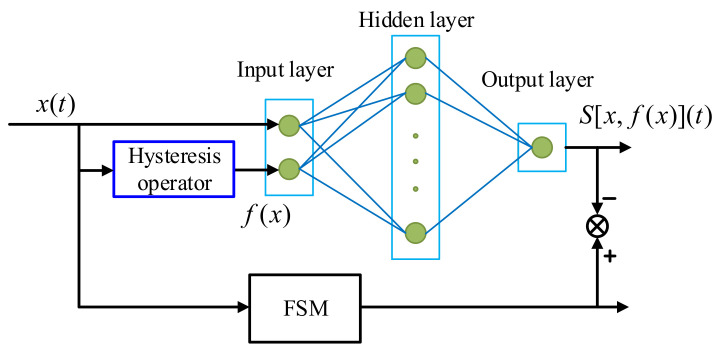
Architecture of the proposed hysteresis model.

**Figure 3 micromachines-12-00732-f003:**
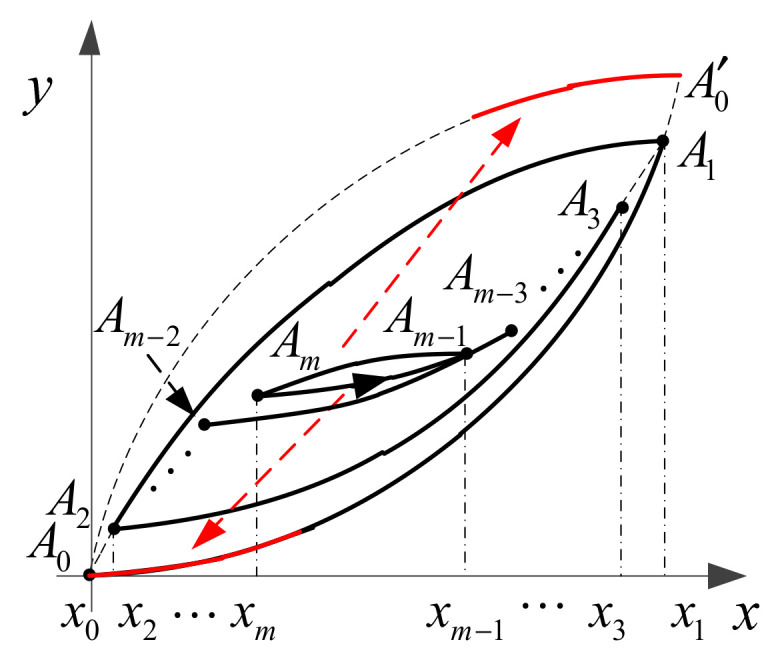
Madelung’s rules: (1) Any hysteresis curve beginning from a turning point is uniquely determined by the coordinates of this point. (2) If any point on the hysteresis curve becomes a new turning point, then the new curve goes back to the previous turning point. (3) If the hysteresis trajectory on the curve $\overset{⏜}{A_{2}A_{1}}$ is continued beyond $A_{1}$, it is transferred to $\overset{⏜}{A_{0}{A^{\prime }}_{0}}$ as if the hysteresis loop $A_{1}A_{2}-A_{2}A_{1}$ does not exist at all.

**Figure 4 micromachines-12-00732-f004:**
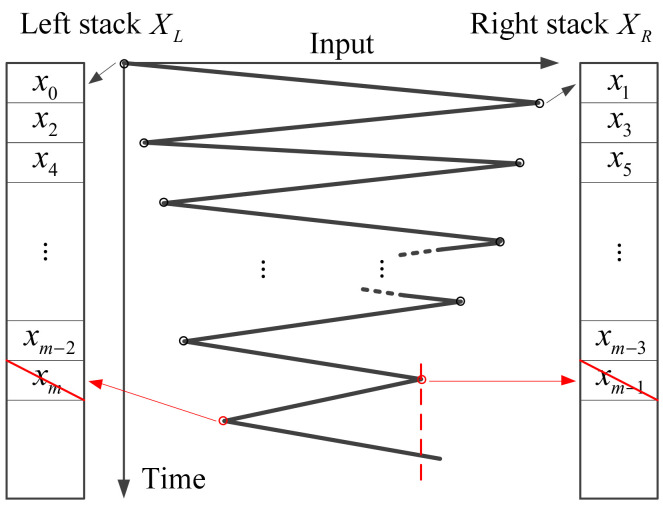
Determination of the current turning point using the wiping-out property.

**Figure 5 micromachines-12-00732-f005:**
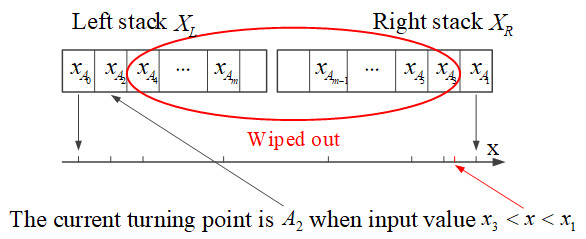
Locate x with the binary search.

**Figure 6 micromachines-12-00732-f006:**
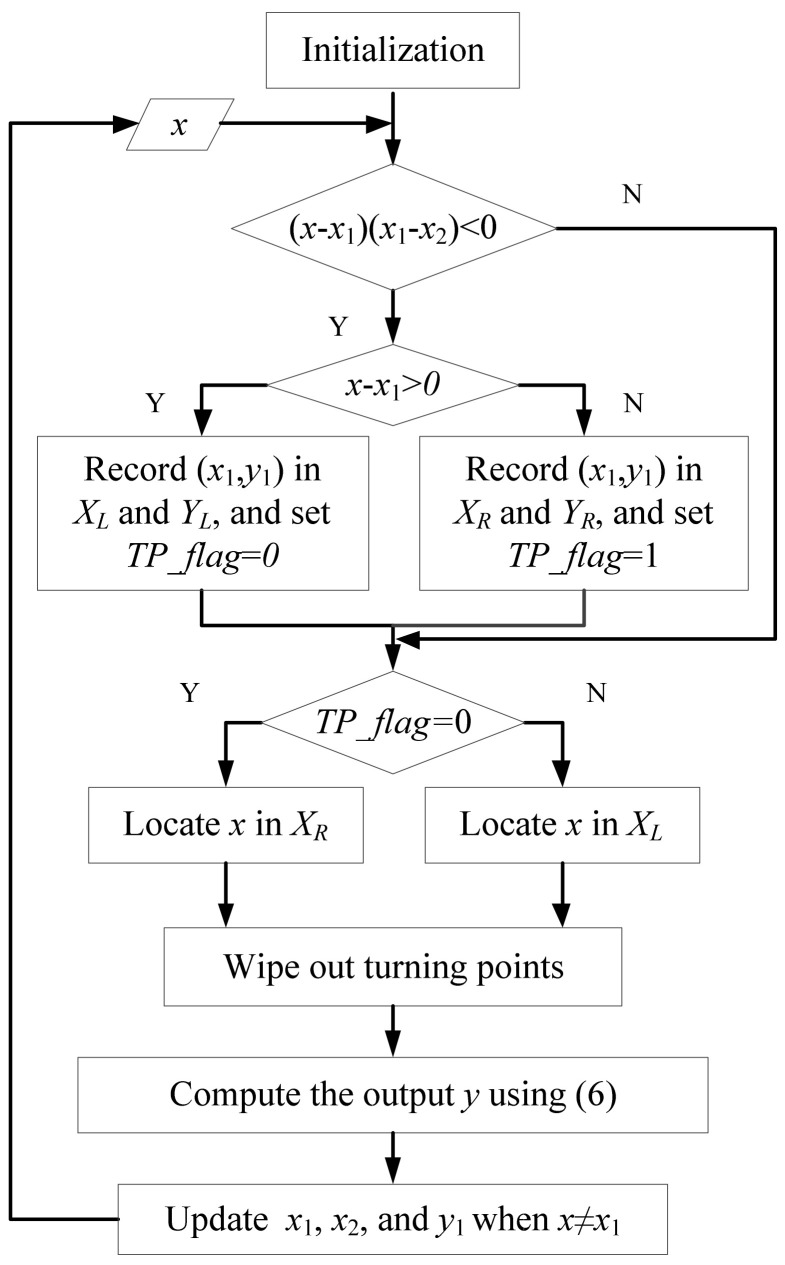
Realization algorithm of the proposed hysteresis operator.

**Figure 7 micromachines-12-00732-f007:**
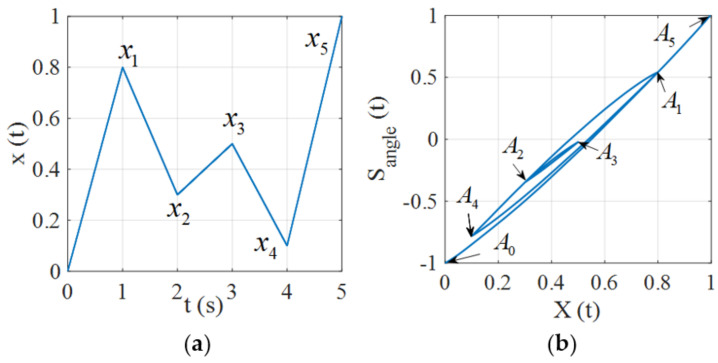
Simulation of the wiping-out property. (**a**) Input signal. (**b**) Hysteresis trajectory.

**Figure 8 micromachines-12-00732-f008:**
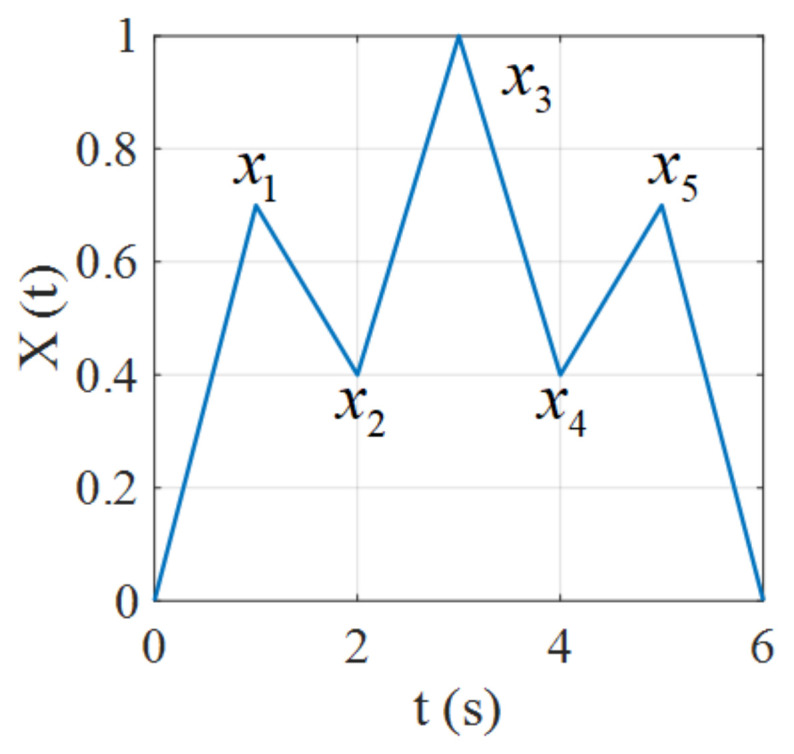
Input signal used in the validation of the congruency property.

**Figure 9 micromachines-12-00732-f009:**
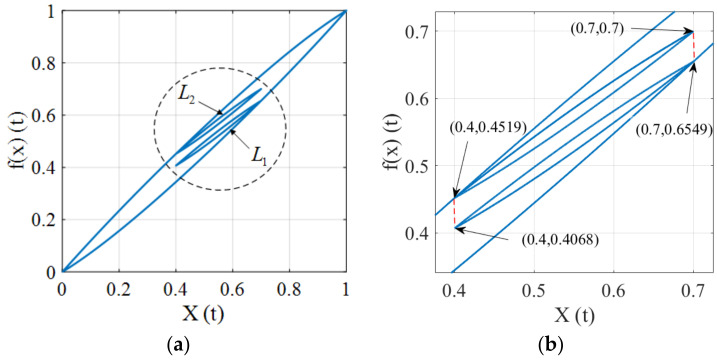
Illustration of the congruency property for the hysteresis operator. (**a**) Hysteresis trajectory. (**b**) An enlarged view of the circle area.

**Figure 10 micromachines-12-00732-f010:**
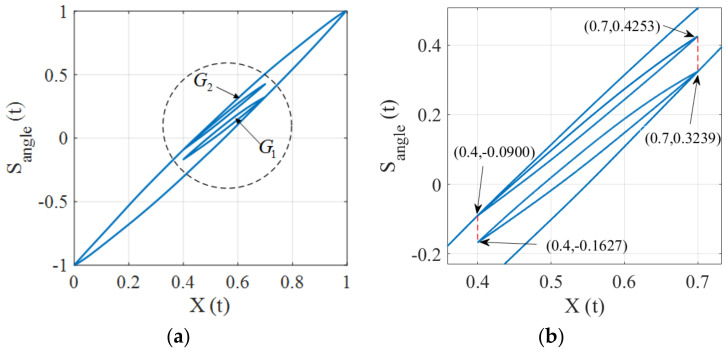
Illustration of the congruency property for the hysteresis model. **(a)** Hysteresis trajectory. **(b)** An enlarged view of the circle area.

**Figure 11 micromachines-12-00732-f011:**
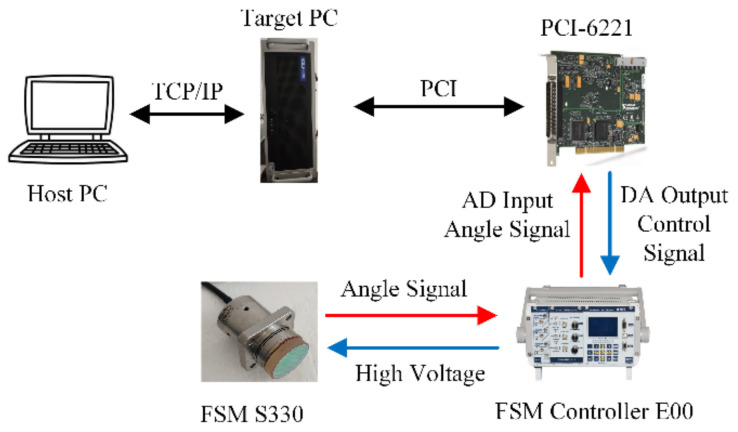
Experiment platform.

**Figure 12 micromachines-12-00732-f012:**
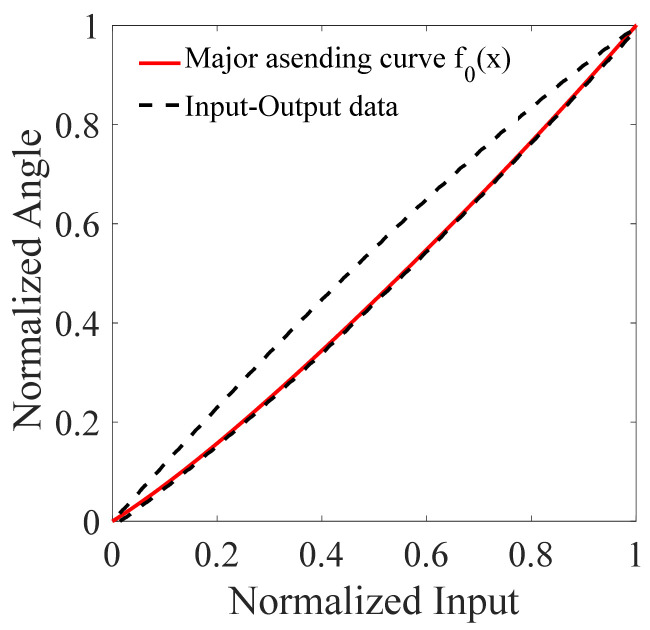
Input-output data and identified major curves.

**Figure 13 micromachines-12-00732-f013:**
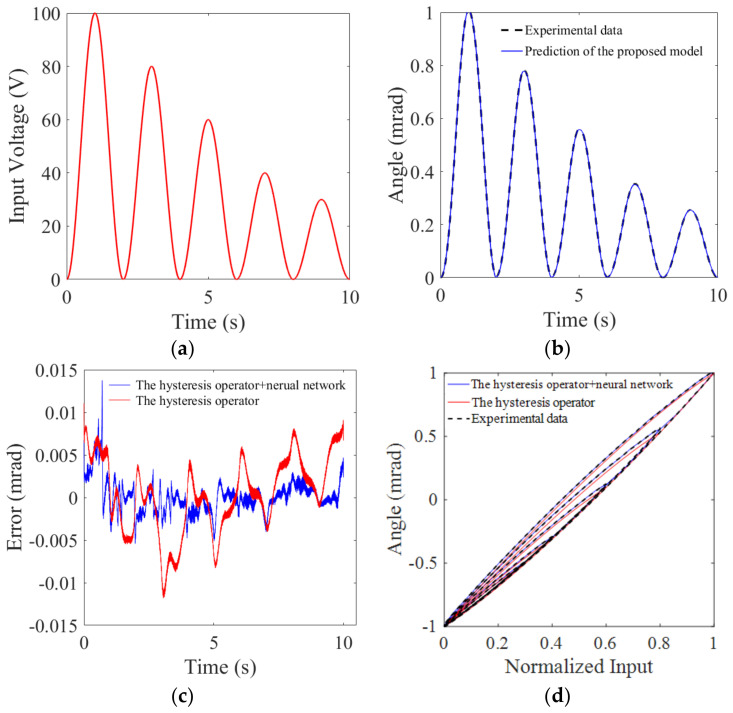
Experimental verification. (**a**) Input signal. (**b**) Actual and predicative angle signals. (**c**) Error. (**d**) Hysteresis loops.

**Figure 14 micromachines-12-00732-f014:**
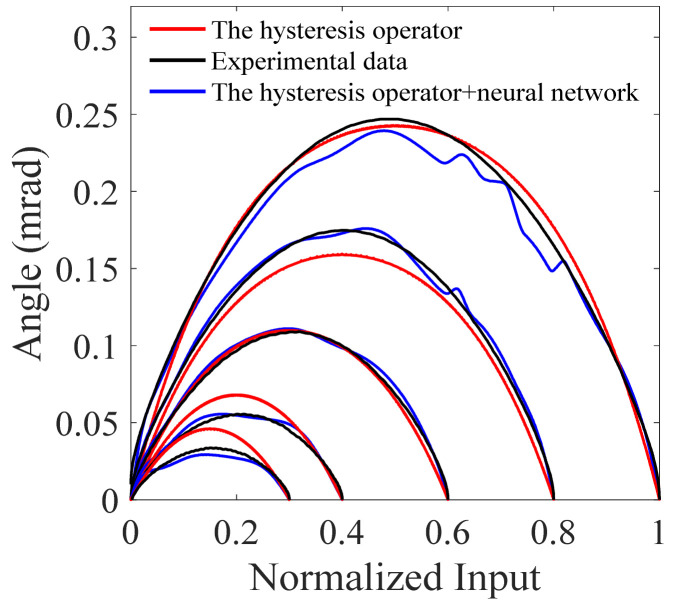
Illustration of the complex nonlinearity of fast steering mirrors.

**Figure 15 micromachines-12-00732-f015:**
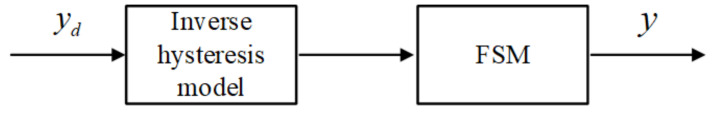
Hysteresis compensation with the inverse hysteresis model.

**Figure 16 micromachines-12-00732-f016:**
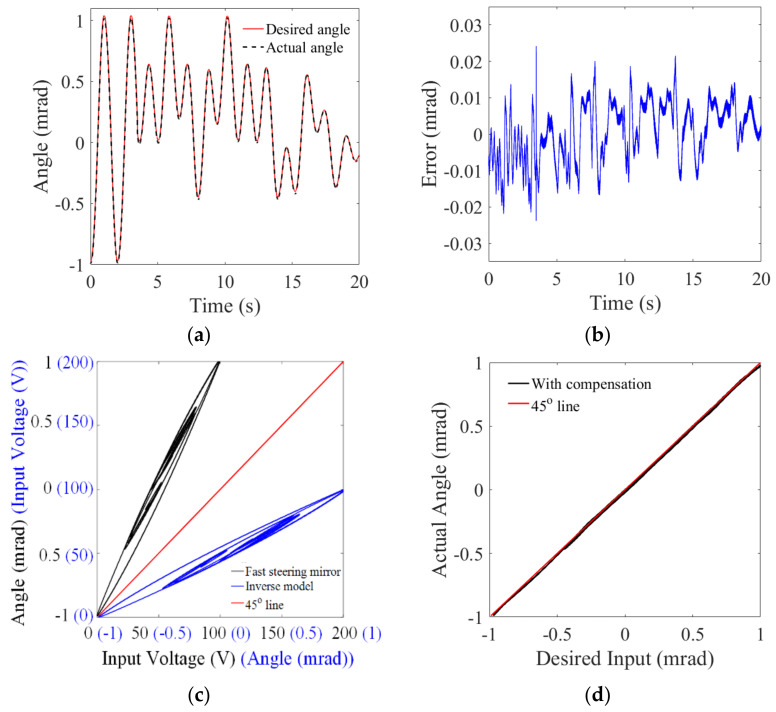
Compensation results. (**a**) Desired and actual angle signals. (**b**) Error. (**c**) Symmetrical relationship between the actual hysteresis and the inverse hysteresis model. (**d**) Actual angle versus input voltage.

**Table 1 micromachines-12-00732-t001:** Parameters of the back propagation neural network.

i	$\left(w_{1i},w_{2i}\right)$	$b_{i}$	$w_{i}$
1	(−3.6257,−3.8904)	6.9822	−0.0531
2	(−74.0750,68.3650)	11.4419	3.1126
3	(2.5646,2.4315)	−3.1634	0.1019
4	(−1.5465,−6.9710)	3.0611	−0.0180
5	(4.6670,8,4231)	3.4183	0.0104
6	(0.1417,−1.9784)	0.0499	−0.5796
7	(−9.3423,12.4904)	−0.2156	−0.0309
8	(1.9804,0.7090)	3.6753	4.8024
9	(7.6728,8,−4.5298)	3.6296	2.2704
10	(0.6836,2.7029)	2.1638	0.1615
11	(−5.0154,2.2743)	−3.6263	7.6830
12	(−1.8607,4.0871)	−3.4273	0.8264

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
