# Peer review of "Hysteresis Modeling and Compensation of Fast Steering Mirrors with Hysteresis Operator Based Back Propagation Neural Networks"

_micromachines, 2021, doi:10.3390/mi12070732_

Round 1

Reviewer 1 Report

The paper is dedicated to an important problem in the field of the fast steering mirrors operation. The presented results show the high efficiency of the proposed model (in the sense of comparison of experimental results and results of numerical simulation).

However, I have some remarks:

  1. It seems more natural to describe the hysteresis Madelung's operator in terms of the corresponding differential equations (see, e.g., M.Krasnosel'skii, A. Pokrovskii Systems with Hysteresis). However, the Authors describe Madelung's operator within the geometric approach and it makes some mandatory suggestions about the form of the hysteresis loop.
  2. To my taste, equation (8) seems at least strange because the Authors do not describe why the coefficients in this polynomial as they are.
  3. In general, hysteresis operators are not reversible because of multivaluedness. Thus, the results of section 4.4 should be described clearly and in detail. Following your description of the inverse hysteresis model, putting two hysteresis operators in series we obtain the input signal as the output of the second hysteresis operator. It seems really strange.  
  4. The Authors should also explain the choice of the type of neural network and its architecture.

In my opinion, the manuscript should be sufficiently revised before acceptance.

Reviewer 2 Report

The manuscript by Kairui Cao and coworkers on „Hysteresis Modeling and Compensation of Fast Steering Mirrors with Hysteresis Operator Based BP neural networks” is a fine study which I find interesting and well executed. I think it deserves publishing unless some minor, but disturbing, English grammar and style errors are corrected and some other small technical issues are attended.

Concerning the language, the manuscript has to be consulted with a native speaker to eliminate cases like:

  • Line 35: Changing the control voltage equals to vary the voltages imposed…
  • Line 48: … hysteresis nonlinearity complex is complex, 48 which is inadequate to be described by…
  • Line 194: … the BP neural network is actually memoryless.
  • Line 232: … where two inner loops the inner loops…

And some others.

The technical issues are the following. They should be corrected as described for the aid to the reader.
1) The meaning of “BP” acronym has not been revealed. In particular it is used in the title

2) The acronym should not be used in figures captions. Instead the full meaning should appear. Figs.1, 4, 5,

3) Except of Figs. 1 and 2, the font of the texts, axes’ labels and legends should be enlarged. They are difficult to read.

4) Fig.3  a) should be Madelung; b) At least some basic description should be given. It is hard to read text on one page and compare with the figure on the other. So, some parts of the description in the text should be placed in the caption. C) The meaning of the red parts of the artwork has to be explained.

5) Line 207: Point A_0 is not marked on Fig. 7.

6) Line 279 and some other places. If (6) and (8) refer to equation, than it is better to write : According to (or using) Eqs. (6) and (8), the ….

7) It is preferable to avoid using acronyms in Conclusion part. It is better to use the full wording.
